# Pharmacogenetic Guidelines for Psychotropic Drugs: Optimizing Prescriptions in Clinical Practice

**DOI:** 10.3390/pharmaceutics15112540

**Published:** 2023-10-27

**Authors:** Antoine Baldacci, Emeric Saguin, Alexander Balcerac, Stéphane Mouchabac, Florian Ferreri, Raphael Gaillard, Marie-Dominique Colas, Hervé Delacour, Alexis Bourla

**Affiliations:** 1Department of Psychiatry, Bégin Army Instruction Hospital, 94160 Saint-Mandé, France; antoinebaldacci.pro@gmail.com (A.B.);; 2Neurology Unit, Percy Army Instruction Hospital, 92141 Clamart, France; 3Department of Psychiatry, Saint-Antoine Hospital, Sorbonne University, 75012 Paris, France; stephane.mouchabac@aphp.fr (S.M.); florian.ferreri@aphp.fr (F.F.); 4ICRIN—Psychiatry (Infrastructure of Clinical Research in Neurosciences—Psychiatry), Brain Institute (ICM), Sorbonne Université, INSERM, CNRS, 75013 Paris, France; 5Department of Psychiatry, Pôle Hospitalo-Universitaire, GHU Paris Psychiatrie & Neurosciences, 75014 Paris, France; r.gaillard@ghu-paris.fr; 6Ecole du Val-de-Grâce, Army Health Service, 75005 Paris, France; marie-dominique.colas@intradef.gouv.fr (M.-D.C.); herve.delacour@intradef.gouv.fr (H.D.); 7Biological Unit, Bégin Army Instruction Hospital, 94160 Saint-Mandé, France; 8Clariane, Medical Strategy and Innovation Department, 75008 Paris, France; 9NeuroStim Psychiatry Practice, 75005 Paris, France

**Keywords:** pharmacogenetics, psychotropic, antidepressant, antipsychotics, prescription assistance, cytochrome

## Abstract

The modalities for prescribing a psychotropic (dose and choice of molecule) are currently unsatisfactory, which can lead to a lack of efficacy of the treatment associated with prolonged exposure of the patient to the symptoms of his or her illness and the side effects of the molecule. In order to improve the quality of treatment prescription, a part of the current biomedical research is dedicated to the development of pharmacogenetic tools for individualized prescription. In this guideline, we will present the genes of interest with level 1 clinical recommendations according to PharmGKB for the two major families of psychotropics: antipsychotics and antidepressants. For antipsychotics, there are *CYP2D6* and *CYP3A4*, and for antidepressants, *CYP2B6*, *CYP2D6*, and *CYP2C19*. The study will focus on describing the role of each gene, presenting the variants that cause functional changes, and discussing the implications for prescriptions in clinical practice.

## 1. Introduction

The prescription of antidepressants (ADs) and antipsychotics (APs) is currently unsatisfactory, often resulting in limited efficacy and an increased risk of side effects. For major depressive disorder (MDD), only 33% of patients achieve remission after their first AD treatment, and this number rises to 67% after trying four different treatments [1]. A study involving 1432 patients on various APs found that 74% had discontinued their medication within 18 months, primarily due to inefficacy or adverse reactions [2].

Beyond the lack of efficacy, potential side effects are also an important issue. Common side effects of ADs include dizziness, nausea, cardiotoxicity, anticholinergic effects, sexual dysfunction, fatigue, and in some cases, weight gain [3]. It is estimated that over 25% of patients on ADs experience these side effects [4]. AP’s adverse events have been widely studied due to their significant impact on patients’ physical health and observance. The main side effects include metabolic syndrome, hyperprolactinemia, QT prolongation, and extrapyramidal symptoms [5].

These adverse effects, coupled with the treatments’ limited efficacy, contribute to a higher mortality rate among patients with severe mental disorders (SMD), such as schizophrenia, other psychotic disorders, bipolar disorder, and moderate to severe depression. Research indicates that SMD patients have a 10 to 20 years reduction in life expectancy compared to the general population [6]. Most deaths among this demographic are attributed to preventable physical illnesses, notably cardiovascular diseases, respiratory diseases, and infections. These individuals are two to three times more likely to die from cardiovascular diseases compared to the general population [6].

Given these statistics, there is a pressing need to refine the prescription methods for ADs and APs. Currently, prescriptions are based on broad recommendations, allowing physicians to choose from a range of molecules across different classes. To adjust dosages, switch medications, or enhance treatment, physicians rely on their understanding of the disease’s progression and the patient’s tolerance to side effects [7]. This approach often results in a period of three to four weeks for ADs and four to six weeks for APs before the efficacy of a drug can be assessed. If a treatment fails, the patient continues to suffer from their illness’s symptoms and potential side effects during this evaluation period.

To improve prescription, current biomedical research is focusing on the development of pharmacogenetic tools for personalized prescriptions [8]. Pharmacogenetics involves considering an individual’s genetic makeup to determine how they metabolize medications. This approach shifts from a one-size-fits-all prescription to a tailored one, where the choice of drug and dosage is based on the patient’s identified genetic variants [9]. This personalized approach accounts for individual variability in both treatment tolerance and efficacy [10].

Earlier studies have delved into the clinical implications of pharmacogenetic tests in managing depression. In 2023, a meta-analysis reviewed randomized controlled trials (RCTs) that used pharmacogenetic tools to assist in prescribing ADs for MDD patients [11]. The analysis revealed that patients whose treatment was guided by pharmacogenetics had a higher remission rate compared to those receiving standard care (OR, 8w 1.58 [1.31 to 1.92]; OR, 12w 1.81 [1.44 to 2.26]). A more recent RCT by “Veterans Affairs” compared AD prescriptions guided by pharmacogenetic tests to standard care. Conducted on 1944 military personnel with MDD, this study found that those guided by pharmacogenetics had a higher 24-week remission rate (OR, 1.28 [95% CI, 1.05 to 1.57]; *p* = 0.02) [12]. Additionally, pharmacogenetics has been shown to reduce treatment side effects [13,14], with a more pronounced impact on side effect reduction than on treatment efficacy.

Studies on antipsychotics are less numerous, but a recent review examines the link between pharmacogenetic variants and outcomes of antipsychotics in patients with schizophrenia spectrum disorders (SSD) [15]. From 2010 to 2022, 29 meta-analyses from 298 studies were analyzed, covering 69 pharmacogenetic variants across 39 genes. Significant effects were observed in pharmacogenetic variants related to antipsychotic response, weight gain, metabolic syndrome, prolactin levels, and other side effects. When analyzing both clinical and preclinical data, in vivo and in vitro studies of aripiprazole emerged with substantial data on the influence of gene variability on its pharmacokinetics and pharmacodynamics, and CYP2D6 metabolizer status is pivotal when administering aripiprazole, supported by FDA’s and EMA’s summary of product characteristics and by DPWG guidelines [16]. Brexpiprazole and cariprazine also have specific pharmacogenetic recommendations, while data on lumateperone and pimavanserin remain limited, and the same applies to other antipsychotics. Finally, some genetic markers that may predict a patient’s likelihood of experiencing side effects or achieving therapeutic benefits from specific antipsychotic drugs have been identified [17]. These findings underscore the potential of pharmacogenetics to tailor antipsychotic treatments to individual patients, potentially improving outcomes and reducing adverse effects. 

In summary, the current prescription methods for antidepressants and antipsychotics present significant challenges in terms of efficacy and side effects, leading to suboptimal patient outcomes. The alarming statistics surrounding the mortality rates of patients with severe mental disorders further emphasize the urgent need for improved prescription strategies. Pharmacogenetics offers a promising avenue for personalized prescriptions, with emerging evidence highlighting its potential in enhancing treatment outcomes and minimizing adverse effects. As we delve deeper into the intricacies of this approach, the following chapters will provide a comprehensive exploration of the theoretical principles underpinning cytochrome pharmacogenetics and a detailed overview of the genetic prescribing guidelines for these critical medications.

## 2. Theoretical Principles of Cytochrome Pharmacogenetics

The pharmacokinetics of drugs, a cornerstone in understanding their therapeutic and adverse effects, is underpinned by the process of drug elimination. This pivotal phase operates through two primary pathways:

***Direct Excretion*:** This pathway pertains to the unaltered excretion of drugs through the renal or biliary systems. A quintessential example of this is the elimination of aminoglycosides, which are excreted in their original form.

***Biotransformation*:** Before excretion, many drugs undergo biotransformation reactions. Specialized enzymes facilitate these reactions, which encompass oxidations and/or conjugations, eventually leading to renal or biliary excretion.

Central to these oxidation reactions is a unique heme-containing protein enzyme, cytochrome P450. Its nomenclature is derived from its distinctive property of binding and absorbing carbon monoxide at a wavelength of 450 nanometers in the visible spectrum [18].

Genes encoding for P450 enzymes, and the enzymes themselves, are designated with the root symbol CYP for the superfamily, followed by a number indicating the gene family, a capital letter indicating the subfamily, and another numeral for the individual gene. The convention is to italicize the name when referring to the gene.

In adults, the metabolism of a plethora of drugs involves three predominant cytochrome families: CYP 1, 2, and 3. These families further branch into subgroups, including CYP1A, CYP2A, CYP2B, CYP2C, CYP2D, CYP2E, and CYP3A, with each playing a distinct role in drug metabolism [19].

Of particular interest are cytochromes CYP2B6, CYP2D6, CYP3A4, and CYP2C19, which exhibit a wide array of genetic variants [20]. These genetic variations significantly influence the metabolism of antidepressant and antipsychotic medications. Intriguingly, most individuals harbor two gene copies responsible for cytochromes CYP2D6, CYP2C19, CYP2B6, and CYP3A4—one allele inherited from the maternal lineage and the other from the paternal lineage. The topic of cytochrome alleles is notably complex, and there are dedicated databases, such as The Human Cytochrome P450 Allele Nomenclature Database of the PharmVar Consortium (https://www.pharmvar.org/, accessed on 20 October 2023), that provide extensive information on this subject. 

### 2.1. From the Genotype to the Phenotype of CYP2D6

The CYP2D6 enzyme showcases a remarkable genetic diversity, boasting over 150 known variants, significantly more than other cytochromes. However, only a handful of these variants exhibit significant allelic frequency. Specifically, for *CYP2D6*, 20 alleles (ranging from *1 to *12, and including *14, *15, *17, *29, *35, *39, *40, *41) account for over 90% of the variants. This distribution varies based on population origins, with the exception of the Sub-Saharan African population, where this panel represents only 77%. Notably, the *149 allele has a 15% allelic frequency in this population, but its activity remains undefined [21]. 

To compute metabolization activity, it is imperative to understand the activity of both alleles, with their cumulative activity representing the overall metabolization activity. The activity value for each CYP2D6 allele is readily accessible on the PharmGKB website [21]. 

The activity score is derived by summing up the activity values of each allele. For instance:
Activity score for CYP2D6 (*1/*2) = 1 + 1 = 2Activity score for CYP2D6 (*4/*4) = 0 + 0 = 0Activity score for CYP2D6 (*3/*9) = 0 + 0.25 = 0.25

Using this activity score, one can assign a phenotype to individuals (Table 1) and different CYP2D6 phenotypes and their percentages in the European population can be listed (Table 2). Phenotypic expressions, reflecting activity scores, were recently re-evaluated during a consensus conference between the two primary scholarly societies overseeing international recommendations: the CPIC (Clinical Pharmacogenetics Implementation Consortium) and the DPWG (Dutch Pharmacogenetics Working Group) [21,22,23]. These phenotypic differences manifest as variations in the rate at which treatments are metabolized by the body. Generally, the faster the rate, the lesser the efficacy (drug eliminated too quickly), and the slower the rate, the higher the risk of side effects (drug accumulation in the bloodstream), especially for treatments requiring cytochromes for transformation into a less active or inactive metabolite.

Conventionally, the different phenotypes are:Ultra-rapid metabolizer (UM) → Increased metabolism speed;Normal metabolizer (NM) → Standard metabolism speed;Intermediate metabolizer (IM) → Reduced metabolism speed;Poor metabolizer (PM) → Slow metabolism speed.

Ultra-rapid metabolizers refer to individuals possessing multiple gene copies encoding for CYP2D6. Instead of the typical two alleles, they might express three or even four alleles. For instance, an individual with a diplotype *1 × 2/*14 has an activity score of:
*1 + *1 + *14 = 1 + 1 + 0.5 = 2.5 (indicating an ultra-rapid profile).

### 2.2. From the Genotype to the Phenotype of CYP2C19

The CYP2C19 enzyme, in contrast to CYP2D6, presents a more limited genetic diversity, with currently identified variants numbering around thirty. The alleles *1, *2, *3, *9, and *17 account for over 90% of these variants, with their distribution varying based on population origins [24].

Given the reduced activity variations among CYP2C19 variants, there are no numerical activity values assigned to each allele of *CYP2C19*. Instead, the PharmGKB website provides a qualitative definition of activity (Table 3). 

Based on this qualitative definition, the CPIC has defined phenotypes (Table 4) according to the diplotype [25]:Ultra-rapid metabolizer (UM) → Highly increased rate of metabolism;Rapid metabolizer (RM) → Increased rate of metabolism;Normal metabolizer (NM) → Normal metabolization rate;Intermediate metabolizer (IM) → Reduced rate of metabolism;Poor metabolizer (PM) → Slow metabolizing rate.

The CPIC has recently introduced nuanced phenotypic definitions to better categorize individuals based on their *CYP2C19* genetic makeup. Two new classifications have been highlighted:

***Likely intermediate metabolizer***: This category encompasses individuals who carry one decreased-function allele and one normal-function allele (*1/*9), or one increased-function allele (*9/*17), or two decreased-function alleles (*9/*9).

***Likely poor metabolizer***: This classification pertains to individuals who possess one decreased-function allele and one no-function allele (*2/*9).

Despite these refined definitions, the therapeutic recommendations for these newly introduced phenotypes remain consistent with those for the previously established “intermediate metabolizer” and “poor metabolizer” categories. Given this overlap in therapeutic guidance, we have opted to streamline our approach by not differentiating between these categories in our recommendations.

### 2.3. From the Genotype to the Phenotype of CYP2B6

The CYP2B6 enzyme, similar to CYP2C19, presents relatively limited genetic diversity. Currently, there are just over 30 identified variants for this enzyme [26]. The alleles *1, *2, *4, *5, *6, *7, *9, *17, and *18 collectively account for more than 95% of these variants. Their distribution, as with other cytochromes, varies based on population origins. However, an exception is observed in the Sub-Saharan African population, where this panel represents only 83% of the population. Similar to those of CYP2C19, CYP2B6 variants exhibit fewer activity variations. As a result, there are no numerical activity values assigned to each allele of *CYP2B6*. Instead, the PharmGKB website provides a qualitative definition of activity (Table 5) and different CYP2B6 phenotypes and their percentages in the European population are provided (Table 6)

Based on this qualitative definition, the CPIC has defined phenotypes (Table 6) according to the diplotype [25]:Ultra-rapid metabolizer (UM) → Highly increased rate of metabolism;Rapid metabolizer (RM) → Increased rate of metabolism;Normal metabolizer (NM) → Normal metabolizing rate;Intermediate metabolizer (IM) → Reduced rate of metabolism;Poor metabolizer (PM) → Slow metabolizing rate.

### 2.4. From the Genotype to the Phenotype of CYP2B6

*CYP3A4* is distinct from other cytochromes in terms of its genetic variations. Remarkably, allele *1 dominates the genetic landscape, representing over 90% of the variants across all populations. Furthermore, *CYP3A4* does not exhibit any alleles leading to increased activity or gene duplication. However, it is worth noting the presence of the *22 allele, which results in reduced cytochrome activity. This allele has an allelic frequency of 5% in the European population [20], implying that 5% of this population exhibits diminished metabolism, different CYP3A4 phenotypes are provided (Table 7). The DPWG has delineated three phenotypes based on the diplotype [27]:
Normal metabolizer (NM) → Standard metabolism speed;Intermediate metabolizer (IM) → Reduced metabolism speed;Poor metabolizer (PM) → Slow metabolism speed.

## 3. An Overview of Genetic Prescribing Guidelines for Antidepressants and Antipsychotics

There are two primary sources for drug prescription recommendations that are based on pharmacogenetics: the Clinical Pharmacogenetics Implementation Consortium (CPIC) (https://cpicpgx.org/ (accessed on 31 July 2023), an international consortium [28], and the Dutch Pharmacogenetics Working Group (DPWG) (https://www.knmp.nl/dossiers/farmacogenetica (accessed on 31 July 2023) [29]. Interestingly, these entities can sometimes offer differing guidelines for the same drug. Several studies have documented the discrepancies between the two groups [30,31,32]. In Canada, the Canadian Pharmacogenomics Network for Drug Safety (CPNDS) has also issued recommendations [32].

The disparities in recommendations stem from differing methodologies. The techniques for grading the level of scientific evidence and the timeliness of the recommendations vary [31]. For instance, the two groups do not update their guidelines in the same year. However, as highlighted in a recent publication, harmonization efforts are underway to establish a consensus between the two entities. This consensus aims to define a unified method for translating the *CYP2D6* genotype to a phenotype [23].

The major regulators, the European Medicines Agency (EMA) in Europe and the Food and Drug Administration (FDA) in the USA, have not issued any specific recommendations. However, the FDA has provided a table listing some specific pharmacogenetic associations [33], indicating groups of patients likely to exhibit altered drug metabolism due to specific genetic variants or inferred phenotypes. However, inclusion in this table does not equate to an FDA endorsement of mandatory pharmacogenetic testing before prescription, unless part of a companion diagnostic. The FDA emphasizes that patient genetics is but one factor affecting the drug response and lacks comprehensive information for safe drug utilization. The FDA itself recognizes that these are not official recommendations, which is why we have not included them in the results, but have summarized them in Table 8, where we have incorporated the FDA’s Table of Pharmacogenetic Associations, to provide a more detailed insight into these recognized gene–drug interactions.

The recommendations and publications from both groups (CPIC, DPWG) are indexed and compiled on the PharmGKB website (https://www.pharmgkb.org/ (accessed on 1 August 2023) [22]. This platform aggregates and disseminates knowledge about the impact of human genetic variations on drug response and tolerance. It also provides an evidence level for each variant–drug interaction. This evidence level is a one-dimensional scale with six possible tiers (refer to Figure 1).

The evidence level is determined based on a score constructed through a five-step process, considering the phenotypic category, *p*-value, cohort size, effect size, and weight [34]. Only evidence levels 1a and 1b come with clinical guidelines that allow the utilization of the interaction information as a prescription aid.

By entering terms like “antidepressant” and “antipsychotics” into the PharmGKB search engine, users are provided with links to prescription guides issued by the CPIC and DPWG [25,27,35,36], as well as all the studied and known genes influencing drug distribution or metabolism.

### 3.1. Antidepressants

When “antidepressant” is entered into PharmGKB, there are 353 clinical annotations and only 50 prescribing information pieces [37]. The only variants with a 1a evidence level are those coding for CYP2B6, CYP2D6, and CYP2C19. All other variants, such as those of ABCB1, have an evidence level ≤ 2, which does not allow expert societies to issue prescription recommendations.

### 3.2. Antipsychotics

Entering “antipsychotics” into PharmGKB yields 305 clinical annotations and merely 24 prescribing information pieces [38]. The only variants with a 1a evidence level are those coding for CYP2D6 and CYP3A4. All other variants, like those of CYP1A2 or ABCB1 or even DRD2, have an evidence level ≤ 2, preventing expert societies from issuing prescription recommendations.

Thus, the most robust findings pertain to genetic variants, or genotypes, of the hepatic cytochrome P450 (CYP) enzymes, such as CYP2B6, CYP2D6, CYP3A4, and CYP2C19. These determine the activity, or phenotypes, of enzymes metabolizing the drug and, consequently, the pharmacokinetics of numerous antidepressants and antipsychotics.

## 4. Summary of Prescription Recommendations Issued by the CPIC for Antidepressants and by DPWG for Antipsychotics

### 4.1. Prescription Recommendations for Antidepressants

Table 9 lists the antidepressants that have prescription recommendations and an evidence grade of 1, along with the associated cytochromes. Tricyclic antidepressants have been grouped together as they do not have differentiated prescriptions and share the same recommendations [35]. For clarity and consistency, we have chosen to rely solely on the CPIC recommendations for antidepressants [25,35]. However, it is worth noting that the DPWG also issues recommendations for antidepressants [36].

Based on these recommendations, a double-entry table was constructed, allowing for prescription advice for all possible phenotype combinations involving CYP2C19 and CYP2D6. An initial version of this table was published in 2022 [39], and Table 9 is an updated version, benefiting from the 2023 update by the CPIC [25]. Moreover, it appears that recommendations can vary depending on the phenotypic profile. For instance, venlafaxine has dose adjustment recommendations for an individual with a poor metabolizer profile for CYP2D6 but has no recommendation for ultra-rapid or intermediate slow phenotypes. For this, we created a gray category “treatment without recommendation”, in which we added commonly used molecules with evidence levels ≤ 2, such as fluoxetine, duloxetine, and mirtazapine.

This revised version adapts existing guidelines and introduces new recommendations, such as those for venlafaxine and vortioxetine. As a reminder, in a 2022 article [39], we based our suggestions for venlafaxine on the DPWG (Dutch Pharmacogenetics Working Group) recommendations [36] and did not issue any recommendations for vortioxetine. For better coherence, this new version of the table (Table 10) relies solely on CPIC recommendations [25,35]. Another significant update pertains to the addition of guidelines for CYP2B6, which is a crucial metabolic pathway for sertraline [25]. Thus, genotyping this cytochrome provides additional information for sertraline. When the genotype information for CYP2D6 is available, Table 11 shows specific recommendations for sertraline. Table 10 provides information on the color code for Table 11 and Table 12.

### 4.2. Prescription Recommendations for Antipsychotics

Table 13 lists the antipsychotics that benefit from prescription recommendations and have a grade 1 level of evidence, along with the associated cytochromes. As a reminder, for the sake of consistency, we have chosen to rely on the DPWG recommendations for antipsychotics, since the CPIC had not yet issued guidelines in 2023 [27].

Based on these recommendations, a double-entry table was constructed, allowing for the provision of prescription advice for all possible phenotype combinations involving CYP3A4 and CYP2D6. Moreover, it appears that recommendations can vary depending on the phenotypic profile. For instance, zuclopenthixol has dose adjustment recommendations for an individual with a slow or intermediate metabolizer profile for CYP2D6 but does not have a recommendation in the case of an ultra-rapid phenotype. For this reason, we created a grey category labelled “treatments without recommendations,” where we added commonly used molecules with a level of evidence ≤2, such as olanzapine and clozapine. It is crucial to note that these two molecules are primarily metabolized by CYP1A2, which currently does not have prescription recommendations. Indeed, most studies on the consequences between the genotypic and phenotypic groups of CYP1A2 did not find a significant pharmacokinetic effect for clozapine and olanzapine. Furthermore, most studies on clinical consequences showed no difference in adverse effects or responses between the different genotypic and phenotypic groups [27]. Table 14 shows the color code and level of evidence for Table 15 that provide antipsychotic prescription recommendations based on CYP2D6 and CYP3A4 phenotypes.

## 5. Discussion

This guide sheds light on recommendations that are backed by a level 1a or 1b evidence base. Currently, only genes encoding for CYP2B6, CYP2D6, CYP2C19, and CYP3A4 meet this criterion. While recommendations by CPIC or DPWG are based on phenotypic expressions, it’s crucial to recognize that these expressions can be influenced by both extrinsic and intrinsic factors. These encompass co-medications, specific foods, certain lifestyle habits, inflammatory diseases, and cancers. When any of these factors interfere with a cytochrome, it can lead to a change in its phenotypic expression, a phenomenon termed “phenoconversion” [40,41]. 

This highlights the challenge posed by the static results provided by pharmacogenetics, which often overlook the interactions between the individual and their environment. Moreover, drug–drug interactions play a pivotal role, especially when introducing new chemical entities to patients. A study revealed that drug-related deaths constituted 7% of all deaths in hospital settings, emphasizing the significance of drug–drug interactions in this context [42]. To elucidate this further, two examples are given:

***Example 1*:** Consider a schizophrenic patient solely treated with risperidone at 6 mg, who develops a severe depressive episode. Following a clinical decision, genotyping of *CYP2D6* and *CYP2C19* is conducted. The results indicate that the patient is a normal metabolizer (NM) for CYP2D6 and an ultra-rapid metabolizer for CYP2C19. Referring to the recommendation table, the decision is made to add paroxetine at 20 mg to the treatment without altering risperidone dosage, as indicated in Table 11. However, it is noteworthy that paroxetine is a potent inhibitor of CYP2D6 (https://drug-interactions.medicine.iu.edu/MainTable.aspx (accessed on 22 August 2023). This induces a phenoconversion of CYP2D6, transitioning the patient from a normal metabolizer to a slow metabolizer [43]. Consequently, the recommendations for prescribing risperidone change. According to the DPWG, for this phenotypic profile, a 50% reduction in the maintenance dose is suggested. Thus, by adding paroxetine, this patient faces a significant risk of risperidone overdose, increasing the likelihood of side effects. An alternative could have been prescribing venlafaxine, which aligns with the patient’s profile and does not cause phenoconversion, or introducing paroxetine while adjusting the risperidone dose downwards. The initial example showcased how a drug can inhibit a cytochrome’s activity. However, it is also feasible to induce this activity.

***Example 2*:** Consider a schizophrenic patient who smokes and is treated with clozapine. Smoking strongly induces the activity of CYP1A2, thereby reducing the blood concentration of clozapine. Hence, to achieve the desired therapeutic effects, this smoking patient requires a higher clozapine dose than if he did not smoke. The primary concern arises when the patient decides to quit smoking. Without the inductive effect of tobacco, the clozapine concentration will suddenly surge, exposing the patient to a high risk of adverse effects. Reductions of clozapine dose by 30% are recommended when a patient on clozapine ceases smoking, as per a meta-analysis that studied the impact of smoking behavior on clozapine blood levels [44]. Such reductions should be guided by a clozapine steady-state trough levels and a thorough clinical risk–benefit evaluation.

Phenoconversion is not a trivial phenomenon. Research from Australia involving 2900 patients revealed significant increases due to phenoconversion. For CYP2D6, the rate of slow metabolizers surged from 5.4% to 24.7%, while for CYP2C19, it escalated from 2.7% to 17%. These figures underscore the importance of considering phenoconversion when prescribing and monitoring drug treatments [45]. Phenoconversion emphasizes the need to transition from a static recommendation to a dynamic one, incorporating patient environment changes that might interfere with drug metabolism [40].

The future of pharmacogenetics lies in dynamic reports based on two information sources and two algorithms to process this information:

***The first information source is genetics:*** What are the patient’s alleles? The first algorithm translates this genetic information into a phenotype, following international recommendations.

***The second information source pertains to the patient’s living environment*:** co-medication, lifestyle, inflammatory disease, cancer. The second algorithm translates this information into an adjusted phenotype, i.e., phenoconversion.

This adjusted phenotype will subsequently assist in prescription.

Algorithms designed to consider a patient’s living environment are under development. One of them, focused on CYP2D6 and accounting only for medications, is already accessible and usable [43] (https://precisionmedicine.ufhealth.org/phenoconversion-calculator/, accessed on 19 October 2023). Phenoconversion rules are also being formulated. Future consensus conferences are anticipated in this regard, as it is vital to define an inhibitor and inducer effect level for each treatment and determine its implication on phenoconversion. Currently, three distinct methods exist to calculate CYP2D6 phenoconversion. Cicali et al. summarize all these emerging issues in their article [43].

While awaiting these consensus conferences and the availability of mature and comprehensive tools, it is incumbent upon the practitioner possessing a cytochrome pharmacogenetics report to be aware of the phenoconversion issue. They must cross-reference their inquiry with the list of substrates, inhibitors, and inducers of the concerned cytochrome. This information is readily available online, with three slightly divergent primary sources [46,47,48].

## 6. Conclusions

This guide underscores the pharmacogenetic recommendations applicable in daily clinical practice to assist in the prescription of antidepressants and antipsychotics. Its aim is to support practitioners equipped with genetic reports for their patients in making treatment choices, considering the patients’ phenotypic expressions. In the discussion section, we emphasized the limitations of these static reports and highlighted the need to transition towards dynamic reports that account for phenoconversion. As mentioned in the introduction, clinical trials [11] assessing the efficacy of these static reports already demonstrate an improvement in patient management in groups guided by pharmacogenetics. Adopting dynamic reports, more aligned with the patient’s metabolic reality, could further enhance these outcomes. Moreover, the number of genes for which recommendations exist is still limited, but this is poised to expand with research advancements. In the future, we might consider integrating other metabolization stages, like the *ABC1* gene, which encodes for Pgp. This protein plays a role in drug absorption at the intestinal level and in the ability of psychotropics to cross the blood–brain barrier [49].

Pharmacogenetics is a rapidly evolving field. Hence, it is vital to consider this swift progression when contemplating conducting meta-analyses based on studies that do not employ the same gene panel or the same recommendations. Future clinical trials, incorporating the concept of phenoconversion, should be subject to separate meta-analyses to re-assess the relevance of pharmacogenetics. Currently, the use of cytochrome pharmacogenetics as a prescription aid tool for psychotropics is not a common practice and is not reimbursed by Social Security in France. It remains primarily confined to a few specialized centers. One reason for this could be the lack of health economics studies, although some have already underscored the potential benefits not only for patient, but also for society at large [50]. It is therefore imperative that future clinical trials systematically incorporate health economics analyses. This could pave the way for a broader adoption of this approach, allowing a larger patient population to benefit from it, even outside a research context.

## Figures and Tables

**Figure 1 pharmaceutics-15-02540-f001:**
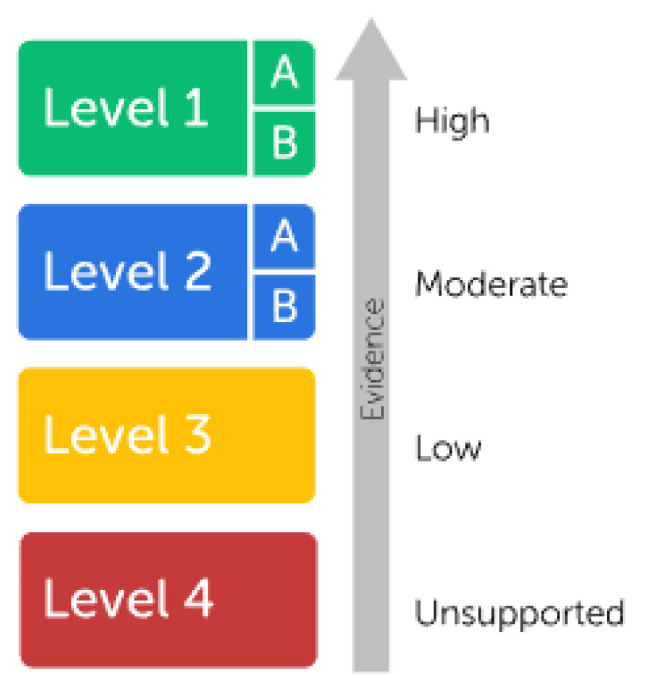
PhamGKB Clinical Annotation Level of Evidence.

**Table 1 pharmaceutics-15-02540-t001:** Activity values for a panel of CYP2D6 variants as per PharmGKB [21,22].

Activity Score	Variants	Allele Type
1	*1, *2, *35, *39	Normal
0.5	*14, *17, *29	Reduced Activity
0.25	*10, *9, *41	Significantly Reduced Activity
0	*3, *4, *5, *6, *7, *8, *11, *12, *15, *40	No Activity

**Table 2 pharmaceutics-15-02540-t002:** Different CYP2D6 phenotypes and their percentages in the European population, adapted from the 2020 consensus conference between CPIC and DPWG [21,23].

Activity Score	Phenotype	Genotype	Example of Diplotype	Phenotype Frequency *
>2.25	UM	Duplications of functional alleles	*1/*1 × N, *1/*2 × N, *2/*2 × N	2%
1.25 to 2.25	NM	Two alleles of normal activity +One of normal activity +One of diminished activity, OR a duplication of normal alleles with an allele of very diminished activity.	*1/*10, *1/*41, *1/*9, *1/*1, *1/*2, *2 × 2/*10, *4/*10	49%
0.25 to 1	IM	Two alleles of diminished activity +Two of very diminished activity +One of normal activity +One of no activity, OR One of diminished activity +One of no activity OR One of very diminished activity +One of no activity OR One of diminished activity +One of very diminished activity.	*4/*41, *10/*10, *41/*41, *10/*41, *41/*41, *1/*5	38%
0	PM	An individual carrying only non-functional alleles	*3/*4, *4/*4, *5/*5, *5/*6	7%

* In the European population; there are 4% of indeterminate phenotypes in the European population. UM: ultra-rapid metabolizer; NM: normal metabolizer; IM: intermediate metabolizer; PM: poor metabolizer.

**Table 3 pharmaceutics-15-02540-t003:** Variant activity for CYP2C19, adapted from PharmGKB [24].

Variants	Allele Type
*17	Increased activity
*1	Normal activity
*9	Reduced activity
*2, *3	No activity

**Table 4 pharmaceutics-15-02540-t004:** Different CYP2C19 phenotypes and their percentages in the European population, adapted from the CPIC guidelines [25].

Phenotype	Genotype	Example of Diplotype	Phenotype Frequency *
UM	Two increased activity alleles	*17/*17	5%
RM	One allele of normal activity +One allele of increased activity	*1/*17	27%
NM	Two alleles of normal activity	*1/*1	40%
IM	One normal function allele +One no function allele	*1/*2, *1/*3, *2/*17	19%
PM	One increased function allele +One no function allele	*2/*2, *2/*3, *3/*3	2%

* In the European population; there are 7% of indeterminate phenotypes in the European population. UM: ultra-rapid metabolizer; RM: rapid metabolizer; NM: normal metabolizer; IM: intermediate metabolizer; PM: poor metabolizer.

**Table 5 pharmaceutics-15-02540-t005:** Variant activity for CYP2B6, adapted from PharmGKB [26].

Variants	Allele Type
*4	Increased activity
*1, *2, *5, *17	Normal activity
*6, *7, *9	Reduced activity
*18	No activity

**Table 6 pharmaceutics-15-02540-t006:** Different CYP2B6 phenotypes and their percentages in the European population, adapted from the CPIC guidelines [25].

Phenotype	Genotype	Example of Diplotype	Phenotype Frequency *
UM	Two increased activity alleles	*4/*4	0%
RM	One allele of normal activity +One allele of increased activity	*1/*4	7%
NM	Two alleles of normal activity	*1/*1	43%
IM	One normal function allele +one decreased function allele OR One normal function allele +One no function allele OR One increased function allele +one decreased function allele OR One increased function allele +One no function allele	*1/*6, *1/*18, *4/*6, *4//18	39%
PM	Two decreased function alleles OR One decreased function allele +One no function allele OR Two no function alleles	*6/*6, *18/*18, *6/*18	8%

* In the European population; there are 3% of indeterminate phenotypes in the European population. UM: ultra-rapid metabolizer; RM: rapid metabolizer; NM: normal metabolizer; IM: intermediate metabolizer; PM: poor metabolizer.

**Table 7 pharmaceutics-15-02540-t007:** Different CYP3A4 phenotypes, adapted from the DPWG guidelines [27].

Phenotype	Genotype	Example of Diplotype
NM	Two alleles of normal activity	*1A/*1A,
		*1B/*1B,
		*1A/*1B,
IM	One allele with normal activity OR One allele with increased activity +One allele with no activity	*1A/*22
PM	Two alleles with no activity	*22/*22

**Table 8 pharmaceutics-15-02540-t008:** FDA’s Table of Pharmacogenetic Associations for Antidepressants and Antipsychotics.

Drug	Gene	Affected Subgroups+	Description of Gene–Drug Interaction
**Pharmacogenetic Associations for which the Data Support Therapeutic Management Recommendations**			
**Aripiprazole**	*CYP2D6*	Poor metabolizers	Results in higher systemic concentrations and higher adverse reaction risk. Dosage adjustment is recommended. Refer to FDA labelling for specific dosing recommendations.
**Atomoxetine**	*CYP2D6*	Poor metabolizers	Results in higher systemic concentrations and higher adverse reaction risk. Adjust titration interval and increase dosage if tolerated. Refer to FDA labelling for specific dosing recommendations.
**Brexpiprazole**	*CYP2D6*	Poor metabolizers	Results in higher systemic concentrations. Dosage adjustment is recommended. Refer to FDA labelling for specific dosing recommendations.
**Citalopram**	*CYP2C19*	Poor metabolizers	Results in higher systemic concentrations and adverse reaction risk (QT prolongation). The maximum recommended dose is 20 mg.
**Clozapine**	*CYP2D6*	Poor metabolizers	Results in higher systemic concentrations. Dosage reductions may be necessary.
**Venlafaxine**	*CYP2D6*	Poor metabolizers	Alters systemic parent drug and metabolite concentrations. Consider dosage reductions.
**Vortioxetine**	*CYP2D6*	Poor metabolizers	Results in higher systemic concentrations. The maximum recommended dose is 10 mg.
**Pharmacogenetic Associations for which the Data Demonstrate a Potential Impact on Pharmacokinetic Properties Only**			
**Amitriptyline**	*CYP2D6*	Ultra-rapid, intermediate, or poor metabolizers	May alter systemic concentrations.
**Amoxapine**	*CYP2D6*	Ultra-rapid, intermediate, or poor metabolizers	May alter systemic concentrations.
**Clomipramine**	*CYP2D6*	Ultra-rapid, intermediate, or poor metabolizers	May alter systemic concentrations.
**Desipramine**	*CYP2D6*	Ultra-rapid, intermediate, or poor metabolizers	May alter systemic concentrations.
**Doxepin**	*CYP2C19*	Intermediate or poor metabolizers	Results in higher systemic concentrations.
**Doxepin**	*CYP2D6*	Ultra-rapid, intermediate, or poor metabolizers	May alter systemic concentrations.
**Escitalopram**	*CYP2C19*	Ultra-rapid, intermediate, or poor metabolizers	May alter systemic concentrations.
**Fluvoxamine**	*CYP2D6*	Poor metabolizers	Results in higher systemic concentrations. Use with caution.
**Imipramine**	*CYP2D6*	Ultra-rapid, intermediate, or poor metabolizers	May alter systemic concentrations.
**Nortriptyline**	*CYP2D6*	Ultra-rapid, intermediate, or poor metabolizers	May alter systemic concentrations.
**Paroxetine**	*CYP2D6*	Ultra-rapid, intermediate, or poor metabolizers	May alter systemic concentrations.
**Risperidone**	*CYP2D6*	Poor metabolizers	Alters systemic parent drug and metabolite concentrations.
**Trimipramine**	*CYP2D6*	Ultra-rapid, intermediate, or poor metabolizers	May alter systemic concentrations.

**Table 9 pharmaceutics-15-02540-t009:** CPIC guidelines for antidepressants with grade 1 PharmGKB recommendations and associated cytochromes variants.

	CYP450		
Antidepressant	CYP2C19	CYP2D6	CYP2B6
CITALOPRAM	●		
ESCITALOPRAM	●		
SERTRALINE	●		●
FLUVOXAMINE		●	
PAROXETINE		●	
TRICYCLIQUES	●	●	
VENLAFAXINE		●	
VORTIOXETINE		●	

**Table 10 pharmaceutics-15-02540-t010:** Color code and level of evidence for Table 11 and Table 12.

Color Code	Level of Evidence	Recommendation
	1A	Drugs that can be used at the standard dosage.
		No recommendation due to a lack of evidence.
	1A	Drugs that can be used with caution.
	1A	drug not recommended for use.

**Table 11 pharmaceutics-15-02540-t011:** Antidepressant prescription recommendations based on CYP2D6 and CYP2C19 phenotypes.

CYP 2C19\2D6	2D6 UM	2D6 NM	2D6 IM	2D6 PM
**2C19 UM**	SERTRALINE	FLUVOXAMINE, PAROXETINE VENLAFAXINE, VORTIOXETINE, SERTRALINE	FLUVOXAMINE, VORTIOXETINE, SERTRALINE	SERTRALINE
	MIRTAZAPINE, DULOXETINE, FLUOXETINE, FLUVOXAMINE, VENLAFAXINE	MIRTAZAPINE, DULOXETINE, FLUOXETINE	MIRTAZAPINE, DULOXETINE, FLUOXETINE, VENLAFAXINE	MIRTAZAPINE, DULOXETINE, FLUOXETINE
	/	/	PAROXETINE (1)	PAROXETINE (2), VORTIOXETINE (2), FLUVOXAMINE (3)
	AMITRIPTYLINE, CLOMIPRAMINE, TRIMIPRAMINE CITALOPRAM, ESCITALOPRAM PAROXETINE, VORTIOXETINE	AMITRIPTYLINE, CLOMIPRAMINE, TRIMIPRAMINE CITALOPRAM, ESCITALOPRAM	AMITRIPTYLINE, CLOMIPRAMINE, TRIMIPRAMINE CITALOPRAM, ESCITALOPRAM	AMITRIPTYLINE, CLOMIPRAMINE, TRIMIPRAMINE CITALOPRAM, ESCITALOPRAM, VENLAFAXINE
**2C19 RM**	SERTRALINE	FLUVOXAMINE, PAROXETINE VENLAFAXINE, VORTIOXETINE, SERTRALINE	FLUVOXAMINE, VORTIOXETINE, SERTRALINE	SERTRALINE
	MIRTAZAPINE, DULOXETINE, FLUOXETINE, FLUVOXAMINE, VENLAFAXINE	MIRTAZAPINE, DULOXETINE, FLUOXETINE	MIRTAZAPINE, DULOXETINE, FLUOXETINE, VENLAFAXINE	MIRTAZAPINE, DULOXETINE, FLUOXETINE
	CITALOPRAM/ESCITALOPRAM (A)	CITALOPRAM/ESCITALOPRAM (A)	PAROXETINE (1), CITALOPRAM/ESCITALOPRAM (A)	CITALOPRAM/ESCITALOPRAM (A), PAROXETINE (2), VORTIOXETINE (2), FLUVOXAMINE (3)
	AMITRIPTYLINE, CLOMIPRAMINE, TRIMIPRAMINE PAROXETINE, VORTIOXETINE	AMITRIPTYLINE, CLOMIPRAMINE, TRIMIPRAMINE	AMITRIPTYLINE, CLOMIPRAMINE, TRIMIPRAMINE	AMITRIPTYLINE, CLOMIPRAMINE, TRIMIPRAMINE, VENLAFAXINE
**2C19 NM**	CITALOPRAM, ESCITALOPRAM, SERTRALINE	AMITRIPTYLINE, CLOMIPRAMINE, TRIMIPRAMINE CITALOPRAM, ESCITALOPRAM, SERTRALINE FLUVOXAMINE, PAROXETINE VENLAFAXINE, VORTIOXETINE	CITALOPRAM, ESCITALOPRAM, SERTRALINE FLUVOXAMINE, PAROXETINE, VORTIOXETINE	CITALOPRAM, ESCITALOPRAM, SERTRALINE
	MIRTAZAPINE, DULOXETINE, FLUOXETINE, FLUVOXAMINE, VENLAFAXINE	MIRTAZAPIN, DULOXETINE, FLUOXETINE	MIRTAZAPINE, DULOXETINE, FLUOXETINE, VENLAFAXINE	MIRTAZAPINE, DULOXETINE, FLUOXETINE
	/	/	PAROXETINE (1), AMITRIPTYLINE/CLOMIPRAMINE/TRIMIPRAMINE (4)	PAROXETINE (2), VORTIOXETINE (2), FLUVOXAMINE (3)
	AMITRIPTYLINE, CLOMIPRAMINE, TRIMIPRAMINE PAROXETINE, VORTIOXETINE	/	/	AMITRIPTYLINE, CLOMIPRAMINE, TRIMIPRAMINE VENLAFAXINE,
**2C19 IM**	/	AMITRIPTYLINE, CLOMIPRAMINE, TRIMIPRAMINE FLUVOXAMINE, PAROXETINE VENLAFAXINE, VORTIOXETINE	FLUVOXAMINE, VORTIOXETINE	/
	MIRTAZAPINE, DULOXETINE, FLUOXETINE, FLUVOXAMINE, VENLAFAXINE	MIRTAZAPINE, DULOXETINE, FLUOXETINE	MIRTAZAPINE, DULOXETINE, FLUOXETINE, VENLAFAXINE	MIRTAZAPINE, DULOXETINE, FLUOXETINE
	CITALOPRAM/ESCITALOPRAM (5), SERTRALINE (5)	CITALOPRAM/ ESCITALOPRAM (5), SERTRALINE (5)	PAROXETINE (1), AMITRIPTYLINE/CLOMIPRAMINE/ TRIMIPRAMINE (4) CITALOPRAM/ESCITALOPRAM (5), SERTRALINE (5)	PAROXETINE (2), VORTIOXETINE (2), FLUVOXAMINE (3), CITALOPRAM/ESCITALOPRAM (5), SERTRALINE (5)
	AMITRIPTYLINE, CLOMIPRAMINE, TRIMIPRAMINE, PAROXETINE, VORTIOXETINE	/	/	AMITRIPTYLINE, CLOMIPRAMINE, TRIMIPRAMINE VENLAFAXINE,
**2C19 PM**	/	FLUVOXAMINE, PAROXETINE VENLAFAXINE, VORTIOXETINE	FLUVOXAMINE, VORTIOXETINE	/
	MIRTAZAPINE, DULOXETINE, FLUOXETINE, FLUVOXAMINE, VENLAFAXINE	MIRTAZAPINE, DULOXETINE, FLUOXETINE	MIRTAZAPINE, DULOXETINE, FLUOXETINE, VENLAFAXINE	MIRTAZAPINE, DULOXETINE, FLUOXETINE
	SERTRALINE (6)	SERTRALINE (6)	PAROXETINE (1), SERTRALINE (6)	PAROXETINE (2), VORTIOXETINE (2), FLUVOXAMINE (3), SERTRALINE (6)
	AMITRIPTYLINE, CLOMIPRAMINE, TRIMIPRAMINE CITALOPRAM, ESCITALOPRAM PAROXETINE, VORTIOXETINE	AMITRIPTYLINE, CLOMIPRAMINE, TRIMIPRAMINE CITALOPRAM, ESCITALOPRAM	AMITRIPTYLINE, CLOMIPRAMINE, TRIMIPRAMINE CITALOPRAM, ESCITALOPRAM	AMITRIPTYLINE, CLOMIPRAMINE, TRIMIPRAMINE CITALOPRAM, ESCITALOPRAM, VENLAFAXINE,

(A) Initiate therapy with recommended starting dose. If the patient does not respond to the recommended maintenance dosing, consider titrating to a higher maintenance dose or switching to an alternative antidepressant not predominantly metabolized by CYP2C19. (1) Consider a lower starting dose and slower titration. (2) Consider a 50% reduction in recommended starting dose, slower titration, and 50% lower maintenance dose. (3) Consider a 25–50% lower starting dose and slower titration or consider an alternative not predominantly metabolized by CYP2D6. (4) Initiate treatment with a 25% reduced dosage and use therapeutic drug monitoring to guide dose adjustments. (5) Initiate treatment with recommended starting dose, consider slower titration and lower maintenance dose. (6) Consider lower starting dose, slower titration and 50% reduction of maintenance dose or consider an alternative not predominantly metabolized by CYP2C19.

**Table 12 pharmaceutics-15-02540-t012:** Prescribing recommendations for sertraline according to CYP2B6 and CYP2C19 phenotypes.

CYP 2C19\2B6	2B6 UM	2B6 NM	2B6 IM	2B6 PM
**2C19 UM/RM**	(a)			
**2C19 NM**			(1)	(2)
**2C19 IM**		(1)	(1)	(3)
**2C19 PM**	(3)	(3)	(3)	

(a) Initiate therapy with recommended starting dose. If the patient does not respond to the recommended maintenance dosing, consider titrating to a higher maintenance dose or switching to an alternative antidepressant not predominantly metabolized by CYP2C19 or CYP2B6. (1) Consider a lower starting dose, slower titration, and lower maintenance dose. (2) Consider a lower starting dose, slower titration, and 25% reduction of standard dose maintenance or consider an alternative not predominantly metabolized by CYP2B6. (3) Consider a lower starting dose, slower titration, and 50% reduction of maintenance dose.

**Table 13 pharmaceutics-15-02540-t013:** DPWG guidelines for antidepressants with grade 1 PharmGKB recommendations and associated cytochromes variants.

	CYP450	
Antipsychotic	CYP2D6	CYP3A4
Aripiprazole	●	
Brexpiprazole	●	
Pimozide	●	
Quetiapine		●
Risperidone	●	
Haloperidol	●	
Zuclopenthixol	●	

**Table 14 pharmaceutics-15-02540-t014:** Color code and level of evidence for Table 15.

Color Code	Level of Evidence	Recommendation
	1A	Molecules that can be used at the standard dosage.
	1A	Molecules that can be used with caution.
	≤2	No recommendation due to a lack of evidence.

**Table 15 pharmaceutics-15-02540-t015:** Antipsychotic prescription recommendations based on CYP2D6 and CYP3A4 phenotypes.

CYP 3A4\2D6	2D6 UM	2D6 NM	2D6 IM	2D6 PM
**3A4 NM**	ARIPIPRAZOLE, BREXIPIPRAZOLE, PIMOZIDE, QUETIAPINE	ARIPIPRAZOLE, BREXIPIPRAZOLE, HALOPERIDOL, PIMOZIDE, QUETIAPINE, ZUCLOPENTHIXOL, RISPERIDONE	ARIPIPRAZOLE, BREXIPIPRAZOLE, HALOPERIDOL, QUETIAPINE, RISPERIDONE	QUETIAPINE
	HALOPERIDOL (a), ZUCLOPENTHIXOL (b), RISPERIDONE (c)	/	PIMOZIDE (4), ZUCLOPENTHIXOL (7)	ARIPIPRAZOLE (1), BREXIPIPRAZOLE (2), HALOPERIDOL (3), PIMOZIDE (5), ZUCLOPENTHIXOL (8), RISPERIDONE (9)
	OLANZAPINE, CLOZAPINE	OLANZAPINE, CLOZAPINE	OLANZAPINE, CLOZAPINE	OLANZAPINE, CLOZAPINE
**3A4 IM**	ARIPIPRAZOLE, BREXIPIPRAZOLE, PIMOZIDE, QUETIAPINE	ARIPIPRAZOLE, BREXIPIPRAZOLE, HALOPERIDOL, PIMOZIDE, QUETIAPINE, ZUCLOPENTHIXOL, RISPERIDONE	ARIPIPRAZOLE, BREXIPIPRAZOLE, HALOPERIDOL, QUETIAPINE, RISPERIDONE	QUETIAPINE
	HALOPERIDOL (a), ZUCLOPENTHIXOL (b), RISPERIDONE (c)	/	PIMOZIDE (4), ZUCLOPENTHIXOL (7)	ARIPIPRAZOLE (1), BREXIPIPRAZOLE (2), HALOPERIDOL (3), PIMOZIDE (5), ZUCLOPENTHIXOL (8), RISPERIDONE (9)
	OLANZAPINE, CLOZAPINE	OLANZAPINE, CLOZAPINE	OLANZAPINE, CLOZAPINE	OLANZAPINE, CLOZAPINE
**3A4 PM**	ARIPIPRAZOLE BREXIPIPRAZOLE, PIMOZIDE	ARIPIPRAZOLE, BREXIPIPRAZOLE, HALOPERIDOL, PIMOZIDE, ZUCLOPENTHIXOL, RISPERIDONE	ARIPIPRAZOLE, BREXIPIPRAZOLE, HALOPERIDOL, RISPERIDONE	/
	HALOPERIDOL (a), ZUCLOPENTHIXOL (b), RISPERIDONE (c), QUETIAPINE (6)	QUETIAPINE (6)	QUETIAPINE (6), PIMOZIDE (4), ZUCLOPENTHIXOL (7)	ARIPIPRAZOLE (1), BREXIPIPRAZOLE (2), HALOPERIDOL (3), PIMOZIDE (5), QUETIAPINE (6), ZUCLOPENTHIXOL (8), RISPERIDONE (9)
	OLANZAPINE, CLOZAPINE	OLANZAPINE, CLOZAPINE	OLANZAPINE, CLOZAPINE	OLANZAPINE, CLOZAPINE

(a) Use 1.5 time the normal dose or consider an alternative (flupentixol, penfluridol, quetiapine, olanzapine, or clozapine). (b) No dose recommendation. If ineffective, try to increase the dose, do not exceed 1.5 times the normal dose. (c) Choose an alternative or titrate the dose according to the maximum dose for the active metabolite (paliperidone): if >15 years old (and >51 kg), consider 12 mg/day, if <15 years old (or <51 kg), consider 6 mg/day: IM 75 mg every 2 weeks. (1) Administer no more than 10 mg/day or 300 mg/month (68–75% of the normal maximum dose of aripiprazole). (2) Use half the normal dose. (3) Use 60% of the normal dose. (4) Use no more than 80% of the normal maximum dose: if >12 years old, up to 16 mg/day, if <12 years old, consider 0.08 mg/kg per day up to a maximum of 3 mg/day. (5) Use no more than 50% of the normal maximum dose: if >12 years old, up to 10 mg/day, if <12 years old, consider 0.05 mg/kg per day up to a maximum of 2 mg/day. (6) For major depressive disorders, consider an alternative, and for other indications, use 30% of the normal dose. (7) Use 75% of the normal dose. (8) Use 50% of the normal dose. (9) Use 67% of the normal dose. If problematic central nervous system side effects occur despite a reduced dose, then reduce the dose further to 50% of the normal dose.

## Data Availability

Data available on demand.

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
