# Peer review of "Pharmacogenetic Guidelines for Psychotropic Drugs: Optimizing Prescriptions in Clinical Practice"

_pharmaceutics, 2023, doi:10.3390/pharmaceutics15112540_

Round 1

Reviewer 1 Report

The authors provide a review of the guidelines for psychotropic drugs.  Overall I do not have significant concerns with the review.  

Minor concerns:

I am unclear why there is a question mark in the title.  Please delete after colon or revise.

Throughout the manuscript contractions are used.  Please write out if according to journal guidelines.  Contractions seem to be less formal English..

Some drugs/medications are in all CAPs, others are capitalized and some are not.  Please revise appropriately.  

Please use HGVS nomenclature and ensure that all genes are in italics and make sure that all proteins do not have an extra space.

It seems redundant to have the phenotypes for each gene after the tables.  Consider revising.

Note that *1 is no variant detected (for most genes).  For people not familiar with the concept, it should be discussed.  For example, page 6, line 184, depending on the test, the percentages of *1 can vary.  

Slow metabolizer is not a CPIC standardized term and should be changed to poor metabolizer, as also indicated in the tables.  

Tables - it is unclear why the phenotype is in parentheses.

The convention for writing genotypes is the smaller allele is written first.  For example, *1/*2, not *2/*1.

All the tables in color are hard to read. Consider changing the font to black in cells with lighter colors.

CPIC is USA-based, but an international consortium.  Please revise line 271 appropriately.

Line 275-the FDA does not provide guidelines.  It provides drug labeling and drug gene association tables.

lines 301 and 307 - it is unclear what is meant by the word "pieces".

line 337- typo - recommendations

The authors use examples of inducers and inhibitors, it would be helpful to know where they are getting this information.  

The authors do not discuss the variability in test content (genes and variants) that is well-documented.  The authors should consider adding this information to the review.  There are efforts to standardize testing.

Most of language seems appropriate except for what was noted above.  

Author Response

Dear Reviewer,

Thank you for your meticulous review and valuable feedback on our manuscript. We have taken the time to address each point of concern you raised. Here is a structured response to your comments:

  1. Title Question Mark: The question mark has been removed as suggested. The title has been changed accordinlgy
  2. Use of Contractions: Following the journal's guidelines, all contractions have been written out. 
  3. Drug/Medication Capitalization: The capitalization has been revised and is now consistent throughout the manuscript as indicated.
  4. HGVS Nomenclature and Italicization: The nomenclature has been revised accordingly, and all genes are now in italics. Additionally, the spacing issue with proteins has been corrected.
  5. Term for Metabolizer: The term "slow metabolizer" has been changed to "poor metabolizer" throughout the manuscript.
  6. Phenotype in Parentheses in Tables: The parentheses have been removed for clarity.
  7. Genotype Writing Convention: The genotypes have been revised so that the smaller allele is written first.
  8. Description of CPIC: The text has been revised to accurately reflect CPIC's international consortium status.
  9. FDA Guidelines Reference: The text has been revised to reflect that the FDA provides drug labeling and drug-gene association tables, not guidelines.
  10. Use of the Word "Pieces": This section was reviewed and revised
  11. Typo on Line 337: The typo has been corrected.
  12. Source of Inducers and Inhibitors Information: We have provided a reference link for the information on inducers and inhibitors. We believe this citation suffices, but are open to revising the bibliography further if needed.
  13. Variability in Test Content: We acknowledge the importance of discussing the variability in test content. Our colleague is looking into this point. If we don't receive the necessary information in time, we may have to omit this, but we understand its relevance.

We have diligently worked to address most of the concerns raised and have made corresponding amendments to the manuscript. We appreciate your thorough review and constructive comments which have significantly contributed to improving the quality of our manuscript. We look forward to any further suggestions you may have.

Best regards,

Reviewer 2 Report

Baldacci and colleagues have provided a solid overview of the current pharmacogenomic information and guidelines available for the prescribing of antidepressants and antipsychotics.  This article will provide a good resource for prescribers, academic teachers as well as medicine and pharmacy students.  It clearly outlines the guidelines available, what they are based on and also provides an insight into the discrepancies and limitations of the field. 

Author Response

Dear Reviewer,

We sincerely appreciate your positive feedback on our manuscript. It is gratifying to learn that you find our work to be a solid overview of the current pharmacogenomic information and guidelines pertaining to the prescribing of antidepressants and antipsychotics. We aimed to create a resource that would be beneficial to prescribers, academic educators, and students in medicine and pharmacy, and your acknowledgment of this is very encouraging.

We are pleased that you found our outline of the available guidelines, their basis, and the discussion on the discrepancies and limitations of the field to be clear and insightful. Your encouraging words serve as a motivation for us to continue contributing valuable information to the scientific community.

Thank you once again for your time and constructive review of our manuscript.

Warmest regards,

Reviewer 3 Report

This article summarizes the guidance for genome-guided antideppresant and antipsychotic treatment.

The authors only focused on the CPIC and DWPG guidelines and did not include the guidance from the major regulators. As such, this is am for shortcoming to make this article useful.

Also, the article misses comparison even between these two research consortia.

Lastly, some of the exhibits could be omitted, eg Figure 1.

Author Response

Dear Reviewer,

Thank you for the time and effort you've invested in reviewing our manuscript. We value your feedback and have taken the time to further investigate the pharmacogenetic guidelines from regulatory bodies as you suggested. We have found that while there are mentions of pharmacogenetic information from regulatory bodies, they do not present the level of detailed guidance comparable to that of CPIC and DWPG.

Specifically, EMA does not provide guidelines and the FDA has provided information for 20 psychiatric medications (antidepressant or antipsychotic) that includes CYP450 pharmacogenetic information, with only 8 of these having specific guidelines for "dosage and administration". Though it's noted that the FDA does recognize other pharmacogenetic associations, the provided information is more limited compared to the extensive and well-structured guidelines provided by CPIC and DWPG. The FDA itself recognizes that various other pharmacogenetic associations exist that are not listed in their Table of Pharmacogenetic Associations. 
Therefore while both the FDA and National Institutes of Health (NIH) have identified pharmacogenetics as key tools in drug development and testing, this acknowledgment does not translate to detailed guidance similar to that provided by CPIC and DWPG which are more comprehensive and practical for clinical implementation.

Moreover, the Clinical Pharmacogenetics Implementation Consortium (CPIC) guidelines have garnered endorsements from reputable professional bodies such as the American Society of Health-System Pharmacists (AHSP) and the American Society of Clinical Pharmacology and Toxicology (ASCPT). This reflects a level of professional acceptance and implementation that stands as a testament to the validity and utility of these guidelines in clinical practice.

Additionally, the Dutch Pharmacogenetics Working Group (DPWG) alongside CPIC have collaborated on clinical guidelines for prescribing psychotropic drugs, indicating a level of international collaboration and recognition. This collaborative effort underscores the global acknowledgment and application of these guidelines, further emphasizing their relevance and importance in the field of psychopharmacology.

Given the aforementioned points, we firmly believe that our focus on CPIC and DWPG guidelines is well-justified and central to providing our readers with a robust and practical understanding of pharmacogenetic guidelines in prescribing psychotropic drugs. We are open to mentioning the limited guidance available from regulatory bodies in the chapter "3. An Overview of Genetic Prescribing Guidelines for Antidepressants and Anti-psychotics" of our manuscript for a well-rounded perspective.

We aim for our manuscript to serve as a reliable and comprehensive resource for clinicians and researchers alike. We look forward to hearing any further suggestions you or the editorial team may have to enhance the quality and relevance of our work.

Best regards,  

Reviewer 4 Report

I have read with interest the paper " Pharmacogenetic Guidelines for Psychotropic Drugs: How to Optimize Prescriptions in Clinical Practice?”. The authors propose a review focusing on a comprehensive study of the theoretical principles underlying cytochrome P450 pharmacogenetics and a detailed description of the guidelines for genetic prescribing of antidepressants and antipsychotics in patients with Severe Mental Disorders. The authors indicate that the purpose of their study is based on the observation that current methods of prescribing antidepressants and antipsychotics have several problems in terms of both efficacy and side effects, leading to suboptimal patient outcomes.

The paper is scientifically accurate and complete. The data presented, processed and tabulated provide useful information for the field. The reference section is adequate, updated and appropriate to back up the points made in the article. The study conveys a clear-cut message. I have no major comments on this article. However, there is a need for the authors to make some changes and implementations.

In the initial part of the paper I would suggest to insert a small paragraph explaining in broad lines the nomenclature of the various alleles of cytochrome P450. This would be useful essentially for readers less familiar with genetics and would make subsequent reading of the paper more understandable.

I also suggest to refer in the text to some databases, which the reader could access and realize the complexity of the allele topic, such as The Human Cytochrome P450 Allele Nomenclature Database of the PharmVar Consortium (http://www.pgrn.org/pharmvar.html).

In this regard I suggest for the same motive to add the links of the PharmGKB, CPIC and DPWG websites in the text.

In Tables 8, 12 add the term VARIANTS (e.g. Table 12: DPWG guideline for antidepressants with grade 1 pharmGKB recommendations and associated cytochromes variants).

Author Response

Dear Reviewer,

We appreciate your constructive feedback on our manuscript. Your positive comments regarding the scientific accuracy, completeness, and usefulness of the data presented are highly encouraging. Following your suggestions, we have made several amendments to improve the clarity and informativeness of our manuscript. Here is a point-by-point response to your comments:

  1. Explanation of Cytochrome P450 Nomenclature:

    • We have added a paragraph in the initial part of the paper explaining the nomenclature of the various alleles of cytochrome P450 to aid readers less familiar with genetics, making the subsequent reading of the paper more understandable. The newly added text describes the designation of genes and enzymes within the cytochrome P450 superfamily, following your suggestion.
  2. Reference to Databases:

    • To provide readers with a resource to delve deeper into the complexity of cytochrome alleles, we have included a reference to The Human Cytochrome P450 Allele Nomenclature Database of the PharmVar Consortium and added the link to the database in the text as suggested.
  3. Links to PharmGKB, CPIC, and DPWG Websites:

    • Following your suggestion, we have added the links to the PharmGKB, CPIC, and DPWG websites in the text to provide readers with additional resources for exploring pharmacogenetic guidelines and allele nomenclature further.
  4. Addition of the Term 'VARIANTS' in Tables 8, 12:

    • We have added the term 'VARIANTS' to the titles of Tables 8 and 12 as suggested, to provide clearer information on the contents of these tables. For example, Table 12 now reads: "DPWG guideline for antidepressants with grade 1 PharmGKB recommendations and associated cytochrome variants."

We hope that these amendments address your concerns and enhance the quality and clarity of our manuscript. We are open to any further suggestions you may have, and we thank you once again for your thoughtful and constructive review.

Warmest regards,

Round 2

Reviewer 3 Report

In this revised version the authors have adequately addressed my previous concerns